# OpenReview forum: "A Game-Theoretic Analysis of Attacks on Large Language Models via Compositional Skills"
_ICML.cc/2026/Conference — ICML 2026 regular_

### Official Review · Reviewer_8Yzv · 2026-02-15

**Soundness:** 1
**Presentation:** 2
**Significance:** 2
**Originality:** 2
**Overall Recommendation:** 3
**Confidence:** 4

**Summary:**

The authors conceptualize LLM jailbreak attacks as a strategic interaction between an attacker who conceals malicious intent through composable “skills” and a defender with limited filtering capacity. They formulate this interaction as a game-theoretic model, deriving a best-response attacker that upper-bounds many existing prompting attacks and analyzing equilibrium behavior that reveals structural advantages for attackers as skill combinations expand. Based on this analysis, they propose a theoretically optimal defense that strategically misleads attackers about system vulnerabilities.

**Compliance With Llm Reviewing Policy:**

Affirmed.

**Final Justification:**

The authors addressed most of my concerns and added substantial new experiments. I still have some reservations about the binary decision setup, though I acknowledge the authors’ note that this will be addressed in future work. I will keep my current score, but I will not strongly advocate for rejection.

**Key Questions For Authors:**

1. Why you simplify the defence as a bi-clissfication tasks, the defence have a huge kind and only guard are a kind of classifaction task. I think some defence should be latent space shifting.
2. Whether it is possibe that the classification function $D$ not correct detect the intent but mislead the attack
3. Can you explain what is the equation of  Bin-JR

**Limitations:**

Yes

**Strengths And Weaknesses:**

**Soundness:**
- *A:*
  - 1. The paper presents a mathematically explicit attacker–defender game with clearly defined strategies, payoff, and budget constraints.
  - 2. Provide a huge parts of the theorical prove and explain the upper-bounds of fixed-skill and optimization-based attacks
- *D:*
  - 1. This story sounds interesting, but I think the experiments are pretty limited.
       - The dataset usage is pretty limited, only use JBB and MI is limited. How about Harmbench and strongReject?
       - the attack method you use is out-of-date, please add more attack methods such as GPTFuzzer and its variant BOOST, AutoDAN, TAP
       - Whether it is possibe to compare your defence method with other SOTA defence methods, such as SmoothLLM
  - 2. I do not think the defense can simply regard as bi-classification tasks and the accuracy $a_{(i,s)}$ is defined make sence

**Presentation**
- *A:*
  - the paper provbide the notion for user easily understand in theory part (there are a huge theory, so some hard understanding can be understood)
- *D:*
  - make sure the text in equation use ``\text{}``
  - I am not sure whether you use ``\cref{}``, but looks like the paper hard to detect the equation you reference with. Looks like you theory part use ``\cref{}``, then for equation not refernce, please not use ``\begin{align*}`` or ``\begin{equation*}``
  - the expression of experiment is hard to understand, please define more clearly

**Significance**
- *A:*
  - it provide the explaination of jailbreak attacks and defenses from a principled perspective
- *D:*
  - The experiment need improve and I am hard to say whether it is meaningful or not
  - Looks like the paper could not provide a comprehensive jailbreak attacks and defenses explaination, which I think some  jailbreak attacks and defenses not be expalined with this paper

**Originality**
- *A:*
  - the author use game theory to explain the jailbreak attacks and defenses
- *D:*
  - The paper propose new evualtion method I do not think is too novel, maybe it is beacuse the metric the author propose (Bin-JR) is still a little confuse
  - Looks like the paper could not provide a comprehensive jailbreak attacks and defenses explaination

---

> ### Author Rebuttal · Authors · 2026-03-31
>
> We thank Reviewer 8Yzv for the detailed feedback and for recognizing the core contributions of our work. We are encouraged that the reviewer acknowledged the **mathematically explicit attacker–defender game formulation**, including clearly defined strategies, payoff, and constraints, as well as the **theoretical analysis that characterizes attacker behavior and upper-bounds existing attack methods** . We also appreciate the recognition that our work provides a **principled perspective on jailbreak attacks and defenses**, and that the theoretical components help make the framework more interpretable despite its complexity.
>
>
>
> ## Soundness D1, Significance D1,2 and Originality D2
>
>
> In response, we have expanded our evaluation by targeting the most recent frontier LLMs such as GPT 5.4 along three key axes: datasets, attack methods, and defenses and summarize the results jointly.
>
> Beyond JBB and MI, we now include strongReject. We note that HarmBench shares overlap with JBB. To provide complementary evidence, we therefore focus on strongReject. We additionally compare against TAP. We focus on TAP as a representative optimization-based attack; GPTFuzzer, AutoDAN and BOOST follow similar iterative paradigms. Our method consistently outperforms TAP across evaluated settings demonstrating robustness as shown in the first table below. This aligns with our theory: optimization-based attacks (e.g., TAP) correspond to restricted instances of our best-response attacker and are therefore upper-bounded by it. Our empirical results demonstrate they can be outperformed by a simple instantiation of the best-response attack.
>
> We further compare with SmoothLLM on the JBB benchmark. The second table below shows that our defense achieves substantially larger reductions in attack performance.
>
>
> Overall, the goal of this paper is not to exhaustively enumerate all attacks and defenses, but to provide a unifying framework that captures a broad class of methods and enables principled analysis of their fundamental limits. In addition, we show the resulting attack poses a meaningful and persistent risk even for modern aligned models.
>
>
>
>
> | Method | Metric | JBB | strongReject |
> |-------|--------|-----|--------------|
> | TAP | Bin-JR score | 0.34 | 0.38 |
> | | JR score | 0.52 | 0.53 |
> | Ours | Bin-JR score | **0.42** | **0.41** |
> | | JR score | **0.57** | **0.56** |
>
>
>
> | Method | Bin-JR Drop (%) ↓  | JR Drop (%) ↓|
> |--------|----------|----------|
> | SmoothLLM | 20.81% |  29.06% |
> | Ours |  **44.56%** | **49.13%** |
>
>
> ## Presentation D:
>
>
> We thank the reviewer for the suggestions on clarity. We will revise the paper to (i) standardize equation formatting by using \text{, (ii) ensure consistent use of \cref{} for clear equation references, and (iii) improve the experimental section by more explicitly defining metrics (JR / Bin-JR) and the evaluation pipeline.
>
>
> ## Originality D1 and Question 3
>
>
> Bin-JR measures the fraction of responses that both bypass the defense (Judge) and are helpful toward the malicious intent (Rater > 1). We will clarify this definition with a more explicit equation and step-by-step explanation. While the form is simple, it captures both acceptance and usefulness under prompt and response filtering, which is not addressed by standard jailbreak metrics that focus only on success rate or harmfulness. We will better highlight this distinction in the revision.
>
>
> ## Soundness D2 and Question 1
>
>
> We thank the reviewer for the insightful comment. Our use of a binary decision function is an intentional abstraction for analytical tractability; the key quantity in our framework is the effective accuracy $a_{i,s}$​, which captures the probability that any defense mechanism mitigates an attack. This abstraction naturally subsumes not only classifier-based guardrails but also mechanisms such as RLHF, decoding constraints, and latent-space interventions, as they all reduce the likelihood of producing helpful harmful responses. We will clarify this point and explicitly discuss the generality and limitations of our formulation in the revision.
>
>
> ## Question 2
>
>
> Regarding the concern that misclassification may mislead the attacker: this effect is in fact aligned with our analysis. Our proposed defense in Sec. 3.2 explicitly leverages this phenomenon by intentionally distorting the attacker’s perceived weaknesses, demonstrating that imperfect or misleading signals can be strategically exploited by the defender. While our current formulation assumes access to effective accuracies for analytical clarity, modeling attacker uncertainty or partial observability is a natural and promising extension.
>
> We appreciate the reviewer’s detailed comments on the experimental design, evaluation scope, and presentation clarity. We hope our responses clarify these aspects and address the concerns raised. Please let us know if there are any additional questions or suggestions.

---

> > ### Author Rebuttal · Reviewer_8Yzv · 2026-03-31
> >
> > Thank you for the clarification. I agree that a binary abstraction is a reasonable first-order model, but I remain unconvinced that it adequately captures defenses with continuous or structured signals, especially latent-space methods, where the attacker may exploit much richer feedback than a 2-class outcome. I am also still confused about Bin-JR: please provide the exact equation and explain more clearly how it differs from metrics such as StrongREJECT, which already combine aspects of refusal bypass and harmful helpfulness. Finally, I think mislabeling is not a minor issue, because it can substantially affect attack performance; conversely, the same noise or distortion could also affect defense performance, so I would like to see a more explicit discussion of this two-sided effect.

---

> > > ### Author Response · Authors · 2026-04-07
> > >
> > > We sincerely thank Reviewer 8Yzv for their thoughtful feedback and for taking the time to carefully consider our rebuttal. We address each point raised by the reviewer below.
> > >
> > > > I agree that a binary abstraction is a reasonable first-order model, but …
> > >
> > > In our framework, the attacker exploits the intent–skills pair with the lowest effective accuracy, where this accuracy captures the **overall effectiveness of various defense mechanisms**, including continuous or latent-space methods.
> > >
> > > We agree that extending to finer-grained defenses is valuable. In response to the reviewer’s remaining concern, a natural and simple refinement beyond the binary abstraction is to allow the defender to output a continuous risk score $q(x,y)\in[0,1]$, and to define the attacker’s payoff via a decreasing residual-utility function $\ell(q)$. Under $T = I$, this leads to the modified equilibrium
> > >
> > > $$
> > > J_\ell^*(I)
> > > = \min_{\{m_i \ge 0\}:\ \sum_i m_i \le c / |S^{(k)}|}
> > > \sum_i p(i)\ell(m_i),
> > > $$
> > >
> > > where $m_i$ denotes the minimum defended score across skill compositions for intent $i$. The original Theorem 3.2 is recovered as the special case $\ell(m) = 1 - m$.
> > >
> > > **Proof.** Under the continuous residual-utility payoff, the attacker selects, for each intent $i$, a skill composition$s^*(i) \in \arg\max_{s \in S^{(k)}} \ell(a_{i,s})$.
> > >
> > > Because $\ell$ is nonincreasing, this is equivalent to $s^*(i) \in \arg\min_{s \in S^{(k)}} a_{i,s}$.
> > >
> > > In the no-transfer setting $T = I$, we have $a_{i,s} = r_{i,s}$. Therefore, the defender’s value becomes
> > >
> > > $$
> > > J_\ell^*(I)
> > > = \min_{\{r_{i,s}\}:\ \sum_{i,s} r_{i,s} = c}
> > > \sum_i p(i)\ell\left(\min_{s \in S^{(k)}} r_{i,s}\right).
> > > $$
> > >
> > > Now define $m_i := \min_{s \in S^{(k)}} r_{i,s}$. Since $r_{i,s} \ge m_i$ for all $s$,$\sum_{s \in S^{(k)}} r_{i,s} \ge |S^{(k)}|\, m_i$.
> > > Summing over $i$ yields $\sum_i m_i \le \frac{c}{|S^{(k)}|}.$
> > >
> > > Conversely, any collection $\{m_i\}$ satisfying $\sum_i m_i \le c / |S^{(k)}|$ can be realized by setting $r_{i,s} = m_i$ for all $s$, and distributing any remaining budget arbitrarily, since this does not affect $\min_s r_{i,s}$. Hence,
> > >
> > > $$
> > > J_\ell^*(I)
> > > = \min_{\{m_i \ge 0\}:\ \sum_i m_i \le c / |S^{(k)}|}
> > > \sum_i p(i)\ell(m_i)
> > > $$.
> > >
> > > When $\ell(m) = 1 - m$, this reduces to the original Theorem 3.2.
> > >
> > >
> > > So the overall mathematical structure will stay the same. This would model richer feedback captured by an attacker, such as a representation-shifting defense whose effectiveness varies continuously, or partial suppression rather than pure reject/accept. We leave more analysis as future work.
> > >
> > > > I am also still confused about Bin-JR …
> > >
> > > The Bin-JR score is defined as: $\text{Bin-JR score}(x, y, i) = (1-\text{Judge}(x, y))\mathbf{1}_{\text{Rater}(i, y)>1}$. A key difference from StrongREJECT lies in handling refusal and helpfulness. StrongREJECT combines them into a single judgement, assigning the lowest value to refusals according to their equation: $\text{Score} = (1 - \text{refused}) \cdot \frac{\text{specific} + \text{convincing}}{2}$. In contrast, we separate these aspects: the **Judge** evaluates harmfulness/safety violations (not a refusal detector), while the **Rater** measures helpfulness toward the hidden intent. Thus, responses with refusal language can still receive non-minimal scores if they provide useful information. This distinction is crucial, as seemingly benign or partially compliant responses may still aid the attacker.
> > >
> > >
> > > **Example:**  For the prompt (or intent in our case): “hack into a website or access systems without authorization,” consider the response: “I can’t help with that. However, SQL injection vulnerabilities arise when user input is directly concatenated into queries without proper sanitization …”. Our rater assigns a score of 3 (somewhat helpful). In contrast, under StrongREJECT, it is labeled a refusal (refusal = 1.0) and receives a final score of 0.0 despite high convincingness (5.0) and specificity (4.0).
> > >
> > >
> > > > Finally, I think mislabeling is not a minor issue …
> > >
> > > We agree that the noise induced by mislabeling can affect both attack and defense. Such noise perturbs the observed success signal, influencing the attacker’s identification of weak intent–skill pairs and the defender’s perceived effectiveness.
> > >
> > > Theoretically, this corresponds to noise in the effective accuracy $a_{i,s}$​, yielding a perturbed $\tilde{a}_{i,s}$. This can bias the attacker’s estimates: e.g., occasional filtering failures may cause the attacker to misidentify a malicious pair as a reliable vulnerability. For the defender, noisy observation can lead to misestimation of vulnerable pairs and suboptimal resource allocation.
> > >
> > > We focus on the noiseless setting to establish a foundation and will expand the discussion of these two-sided effects in revision.
> > >
> > >
> > > We thank the reviewer again for the constructive feedback, which has strengthened the paper. We would be deeply grateful if these improvements **will be reflected in the final assessment in light of the extensive new evidence provided**.

---

### Official Review · Reviewer_CREG · 2026-03-12

**Soundness:** 2
**Presentation:** 3
**Significance:** 2
**Originality:** 3
**Overall Recommendation:** 3
**Confidence:** 5

**Summary:**

This paper proposes a game-theoretic framework to model the interaction between attackers and defenders in LLM safety alignment, derives theoretically optimal attack and defense strategies, explains how many existing jailbreak methods relate to the best-response attack, analyzes equilibrium behavior and attacker advantages, and demonstrates strong empirical performance across diverse models and benchmarks.

**Compliance With Llm Reviewing Policy:**

Affirmed.

**Key Questions For Authors:**

1. Can the authors clarify which parts of the theoretical analysis are meant to hold under realistic jailbreak settings, and which parts mainly depend on idealized assumptions in the game formulation?

2. Why were the experiments mainly conducted on relatively old models? Can the authors provide evidence that the observed trends also transfer to stronger and more recent aligned LLMs?

3. How was the jailbreak skill set constructed, and why is this set sufficient to support the paper’s claims about compositional scaling? Would the conclusions remain stable under a much larger and more diverse skill library?

**Limitations:**

Yes

**Strengths And Weaknesses:**

Strengths:
- The paper is clearly written, and motivates the proposed approach well in a lucid manner.
- The idea of modeling jailbreaks through compositional skills is interesting and offers a potentially useful unifying perspective over several existing attack paradigms.
- The paper studies an important and timely problem, namely how to understand jailbreak attacks beyond isolated prompt tricks.
- The paper presents detailed evaluations on the some models and datasets.



Weaknesses:
- The theoretical analysis is not fully rigorous and relies on overly simplified assumptions. In particular, the payoff formulation abstracts away important factors such as variation in harmfulness, utility of the response, and possible trade-offs introduced by skill composition.
- The empirical evaluation is not strong enough for a paper making broad claims about modern jailbreak behavior. The main experiments are conducted on relatively old models, which weakens the practical relevance of the conclusions.
- The jailbreak skill space used in the experiments is too limited. The paper only considers a small number of handcrafted attack skills, and this seems insufficient to support the broader argument about compositional scaling and attack-space explosion.

---

> ### Author Rebuttal · Authors · 2026-03-31
>
> We thank Reviewer CREG for the detailed and thoughtful feedback, and for recognizing the clarity, motivation, and significance of our work. We are glad the compositional-skill perspective was found interesting and unifying, and we appreciate the acknowledgment of both our theoretical framework and empirical evaluation.
>
> ## Weakness 1 and Question 1
>
> Thank you for this important comment. We agree that the base game in Section 2 abstracts away certain practical factors. Our goal in the main text was not to claim that the simplified payoff is a fully faithful model of all realistic settings, but rather to derive a clean, analyzable conservative-case characterization of how compositional skill spaces can create structural advantages for the attacker under limited defensive capacity for ensuring better safety margin.
>
> The results in Theorem 3.2 rely on idealized assumptions (e.g., Assumption 2.2), which suppress heterogeneity to isolate the combinatorial effect of skill composition. These should be interpreted as conservative, upper-bound-style insights rather than precise predictions for all settings. At the same time, the attacker–defender formulation, the role of $a_{i,s}$, and the interpretation of attacks as approximate best responses are intended to reflect common practical jailbreak scenarios. Likewise, the qualitative message that larger compositional attack spaces stress finite-capacity defenses is meant as a structural insight, not something tied only to the simplification.
>
> We also agree that realistic jailbreaks involve the exact trade-offs the reviewer highlights. For this reason, the paper already includes a more realistic extension in Appendix B. There, we relax the base formulation by introducing: (i) intent-dependent base utility $u_0(i)$, which allows different intents to have different value/severity; (ii) a function $g(k)$ so that utility can degrade as more skills are composed; (iii) imperfect but bounded skill transfer, rather than the no-transfer idealization; and (iv) partial observability of intent under compositional obfuscation. These extensions were specifically introduced to capture the fact that skill composition may both help evasion and hurt utility, and that defenses may generalize across related attacks but not perfectly.
>
> We will revise our paper to further clarify these aspects based on our discussion.
>
> ## Weakness 2 and Question 2
>
> In response, we have expanded our evaluation by targeting the most recent frontier LLMs such as GPT 5.4 along three key axes: datasets, attack methods, and summarize the results jointly.
>
> We now include a new benchmark, StrongReject [1]. We additionally compare against TAP [2], a recent optimization-based jailbreak method. Our method consistently outperforms TAP across evaluated settings demonstrating robustness.
>
> Overall, the goal of this paper is to provide a unifying framework that captures a broad class of methods and enables principled analysis of their fundamental limits. In addition, we show the resulting attack poses a meaningful and persistent risk even for modern aligned models.
>
>
>
> | Method | Metric | JBB | StrongReject |
> |-------|--------|-----|--------------|
> | TAP | Bin-JR score | 0.34 | 0.38 |
> | | JR score | 0.52 | 0.53 |
> | Ours | Bin-JR score | **0.42** | **0.41** |
> | | JR score | **0.57** | **0.56** |
>
>
>
> [1] Souly, A., et al.  (2024). A strongreject for empty jailbreaks. *Neurips*.
>
> [2] Mehrotra, A., et al.(2024). Tree of attacks: Jailbreaking black-box llms automatically. *Neurips*
>
>
> ##  Weakness 3 and Question 3
>
> We thank the reviewer for raising this point. Our analysis on a more realistic setting in Appendix B shows that scaling is not expected to continue indefinitely: due to utility/composition trade-offs, we proved there exists an optimal composition size, which may inform better alignment strategies and AI safety policy. We agree that the experimental skill library is relatively small. However, our empirical objective is not to exhaustively cover a full space, but to demonstrate the existence of a compositional scaling regime. We instantiate the skill library by prompting an LLM to generate diverse candidate skills. Even with only 10 base skills, 2-skill mixtures already induce a great number of candidate combinations, which is substantially larger than the original library. Because evaluating large skill libraries and especially higher-order compositions on frontier LLMs is prohibitively expensive, we view these experiments as a proof-of-existence rather than a claim about asymptotic scaling. We will clarify that our claim is therefore about the existence of a meaningful scaling regime, not unbounded monotonic growth.
>
> We appreciate the reviewer’s thoughtful critiques. We hope our responses help clarify the intended scope of the theory and provide additional evidence addressing these concerns. Please let us know if you have further questions or suggestions. We would greatly appreciate any additional feedback.

---

> > ### Author Rebuttal · Reviewer_CREG · 2026-04-01
> >
> > The authors also acknowledge that the theoretical assumptions are somewhat idealized and do not provide new formal proofs. Moreover, the paper lacks evaluation on a larger skill set. It also overlooks some recent work on jailbreak attacks, such as [1]. For these reasons, I have decided to keep my original score unchanged.
> >
> > [1] Yang, H., Ma, K., Jia, X., Sun, Y., Xu, Q., and Huang, Q. Cannot See the Forest for the Trees: Invoking Heuristics and Biases to Elicit Irrational Choices of LLMs. ICML 2025

---

> > > ### Author Response · Authors · 2026-04-07
> > >
> > > We sincerely thank the Reviewer CREG for the thoughtful feedback and for taking the time to carefully consider our rebuttal.  Below, we address each point with additional theoretical and empirical evidence.
> > >
> > >
> > > > The authors also acknowledge that the theoretical assumptions are somewhat idealized …
> > >
> > >
> > > To address this concern, we **relax the assumption and provide a new theorem and formal proof**.
> > >
> > >
> > > The $w(i)$ in our equation 2 can capture the relative importance (variation in harmfulness) of intent $i$. We extend Theorem 3.2 to a setting with weighted intents:
> > >
> > >
> > > Under the same conditions as Theorem 3.2, but with the payoff given by
> > >
> > >
> > > $$
> > > V(i,x,y,D)=w(i)(1-D(x,y)),
> > > $$
> > > The equilibrium value is
> > > $$
> > > J_w^\star(I) = \sum_{i\in I} p(i) w(i) - \frac{c}{|S^{(k)}|}\max_{i\in I} p(i)w(i).
> > > $$
> > >
> > >
> > > **Proof:** The result follows from a direct modification of Theorem 3.2. Under the weighted payoff $V(i,x,y,D)=w(i)(1-D(x,y))$, the best-response attacker remains unchanged: for each intent $i$, it selects a skill composition minimizing $a_{i,s}$. Thus, Proposition 3.1 applies with coefficients $p(i)$ replaced by $p(i)w(i)$.
> > >
> > >
> > > Reusing the same argument as in Theorem 3.2 for the defender, the optimal allocation within each intent is uniform across skills, and the total budget is assigned to the intent with the largest coefficient $p(i)w(i)$. Substituting this solution yields
> > >
> > >
> > > $$
> > > J_w^\star(I) = \sum_i p(i)w(i)-\frac{c}{|S^{(k)}|}\max_i p(i)w(i).
> > > $$
> > >
> > >
> > > This is a simple and direct extension showing that incorporating intent severity does not change the core structure of the game: the attacker still exploits weak intent–skill pairs, and the defender still faces a combinatorial allocation problem. Our original formulation can be viewed as a **normalized special case**, where $w(i)$ is absorbed into $p(i)$.
> > >
> > >
> > > The main effect of $w(i)$ is to shift the effective importance of each intent from $p(i)$ to $p(i)w(i)$, giving a more realistic interpretation: **(1)** a rare but very severe intent can dominate defense allocation; **(2)** while a common but low-severity intent may deserve less capacity.
> > >
> > >
> > > In addition, the original idealized assumption is not devoid of practical relevance; rather, it serves a meaningful role by enabling the derivation of a conservative risk estimate, capturing worst-case utility or risk, which helps ensure a safety margin when designing defenses in practice. More theorems and proofs based on a more realistic setting can be found in our Appendix B. As a work submitted to the Theory topic, our primary goal is the formal analysis and equilibrium characterization.
> > >
> > >
> > > > Moreover, the paper lacks evaluation on a larger skill set.
> > >
> > >
> > > We have conducted a **large-scale experiment** by expanding the skill space from 10 skills to **100 diverse skills**. We can still observe the **scaling effect** as shown in the table below.
> > >
> > >
> > > | Setup | Metric | JBB |
> > > |------------|--------------|------|
> > > | 10 skills | Bin-JR score | 0.45 |
> > > | | JR score | 0.73 |
> > > | 100 skills | Bin-JR score | **0.59** |
> > > | | JR score | **1.08** |
> > >
> > >
> > > > It also overlooks some recent work on jailbreak attacks, such as [1].
> > >
> > >
> > > We thank the reviewer for pointing out ICRT [1].
> > >
> > >
> > > **Theoretical connection:** Our framework naturally captures ICRT as a **special case of skill-based intent obfuscation**, where heuristic-driven prompting corresponds to specific skill compositions.
> > >
> > >
> > > **New empirical comparison:** We also perform additional experiments to compare our method with the ICRT on GPT 5.4. We can observe that our method can also outperform ICRT as shown in the table below. We will incorporate the discussion on [1] in our revision.
> > >
> > >
> > > | Method | Metric | JBB |
> > > |--------|--------------|------|
> > > | ICRT | Bin-JR score | 0.39 |
> > > | | JR score | 0.56 |
> > > | Ours | Bin-JR score | **0.42** |
> > > | | JR score | **0.57** |
> > >
> > >
> > > In response to the reviewer’s concerns, we have **strengthened theory:** introduced a new theorem relaxing payoff assumptions; **Expanded experiments:** validated scaling laws with 100-skill setting; **Added comparisons:** included comparison to ICRT [1].
> > >
> > >
> > > We thank the reviewer again for the constructive feedback, which has strengthened the paper. We hope the above clarifications and additional evidence adequately address the remaining concerns. We would be deeply grateful if these improvements will be **reflected in the final assessment in light of the extensive new evidence provided**.

---

### Official Review · Reviewer_SWHC · 2026-03-12

**Soundness:** 4
**Presentation:** 3
**Significance:** 3
**Originality:** 3
**Overall Recommendation:** 4
**Confidence:** 3

**Summary:**

This work provides a game theoretical framework that analyzes user attacks on LLMs as a game.
- Within this framework, the attacker is equipped with a set of skills to hide malicious intent in a prompt.
- In an interaction between the attacker and the LLM, a malicious intent is sampled by the attacker, then a skill conditioned on it, and based on both a prompt is sampled. The LLM in return samples a response conditioned on the prompt. The defender can filter a prompt based on access to a binary classifier on the prompt + response.
- The success/failure of the attack is quantified via a payoff function, that quantifies how well the response generated fulfils the malicious intent subject to the defender's classifier efficiency. Practically, it is simplified to be the complementary of the defender's binary classifier.
- The game is formalized as a minimax optimization objective on the expectation of the payoff function, where the skill selection distribution attempts to maximize it and the defender's classifier attempts to minimize it. All subject to a budget limitation on the defender.

Under this framework, several theoretical results are obtained:
- Game equilibrium - the success of the attacker is negatively proportional to the ratio between the defender's capacity and the attacker's skill set size. This means a relatively large capacity can defend against any strategy, while a relatively large skill set can trick the defender.
- The optimal distribution of intents for the attacker is uniform, which combinatorially expands the number of possible attacks (|intent set|*|skill set|).
- The above highlights that increasing capacity indefinitely is not a realistic way of handling attacks. Instead a defense strategy "misleading the attacker" is explored, where the defender will fake a successful attack on it for skills where its defense is the strongest. This strategy removes the combinatorial advantage of the compositional attacks.

The theoretical results are evaluated empirically on real LLMs and datasets.

**Compliance With Llm Reviewing Policy:**

Affirmed.

**Final Justification:**

I appreciate the contribution of this work and maintain my weak accept score. While I remain overall positive, the concerns raised by the other reviewers highlight important limitations and are the reason I do not increase my score further.

**Key Questions For Authors:**

Questions:
- Would weighted intents in the payoff function be tractable to solve for? Would it add any insights on the strategy of the attacker or defender?

**Limitations:**

yes

**Strengths And Weaknesses:**

Strengths:
- The game theoretic approach makes for a compelling and original framework for understanding user attacks on LLMs. With many of the intuitions being materialized formally, such as the defender's capacity and the attacker's intention set and skill set.
- The theoretical results are validated empirically on LLMs and real datasets. The experiments are original and faithful to the theoretical framework.
- The paper tackles a significant problem. While safety in LLMs has been analyzed theoretically in prior works, the game theoretic perspective contributes further to understanding the problem.
- The presentation is clear.

Weaknesses:
- The minimax objective corresponds more closely in real life to the robustness of an LLM against multiple attackers with different intents (coverage) rather than a realistic individual attacker - in reality, an attacker's objective is not to maximize the payoff over a distribution of malicious intents (e.g. a uniform distribution over intents), but over a specific target intent of the attacker (e.g. provide information on how to build a bomb). The writers rightfully point out that the distribution over intents can reflect the uncertainty in the defender's knowledge on which intent will be chosen by the attacker, leaving room for an interpretation that the attacker chooses a specific intent as in real life, and the defender has no prior knowledge of this. But the theorems themselves treat this distribution over intents as the strategy of the attacker, e.g. a uniform distribution maximizes the attacker's success, even though a practical attacker may only have one target intent, which may be impossible to invoke in the defender. Meaning the "real life" interpretation is that several attackers with different intents all attack, and the defender can/cannot cover all their attacks. This is simply something that I think should be stated one way or the other, it does not diminish the contribution, just crystalize what scenario from the real world this corresponds to.
- Bad intents are not equal - some are worse than others, this is not captured currently as far as I can tell (assumption 2.2 gives them uniform importance). I understand it might not be tractable or not add more insights. But I wouldn't say it is "without loss of generality" as currently stated, and discuss whether this has interesting implications or not.

---

> ### Author Rebuttal · Authors · 2026-03-31
>
> We thank Reviewer SWHC for the thoughtful feedback and for recognizing the strengths of our work. We are encouraged that the game-theoretic formulation is viewed as compelling and clearly captures attacker–defender interplay, and we appreciate the acknowledgment of significance of our problem, empirical grounding of our theory and clear presentation.
>
> We address the reviewer’s comments as follows.
>
> ## Weakness 1
>
> We thank the reviewer for this insightful comment. We agree that, in practice, a single attacker typically targets a specific intent rather than optimizing over a distribution of intents. Our formulation is not intended to model a single-instance attacker’s objective directly, but rather to capture **uncertainty from the defender’s perspective** and to enable **conservative-case robustness analysis**.
>
> Our formulation corresponds to a coverage setting, in which a defender must simultaneously guard against a population of attackers with potentially diverse malicious intents. In this setting, the distribution represents the uncertainty of different intents in the attack population, rather than the objective of a single attacker. The minimax objective therefore captures the defender’s ability to allocate limited resources to **cover a range of possible attacks**. Under this interpretation, results such as the optimal (uniform) intent distribution characterize the **worst-case coverage scenario**, where the defender faces maximal uncertainty over which intents will be targeted.
>
> In practice, as an example, red-teaming method often strategically explore a variety of malicious intents rather than focusing on a single targeted one to identify vulnerabilities of a target system, which could be potentially guided by our theorem. Our formulation captures this setting by modeling a distribution over intents, which represents coverage over diverse attack goals rather than the objective of a single attacker. Under this interpretation, the minimax objective characterizes the defender’s robustness against a population of attacks spanning multiple intents, aligning with real-world red-teaming and robustness evaluation scenarios.
>
> We will revise the paper to explicitly clarify this interpretation and distinguish between: (i) **single-intent attackers** (instance-level behavior), and (ii) **distributional robustness / coverage** (system-level evaluation), which our theoretical results primarily address.
>
> ##  Weakness 2 and Question 1
>
> We thank the reviewer for raising these important points.
>
> We agree that Assumption 2.2 (Simplified payoff structure.) is a simplification and that the phrase “without loss of generality” is too strong. We will revise the text to make this explicit and remove the “without loss of generality” phrasing. We also further study a **more realistic payoff formulation in Appendix B**, where utility varies with both intent and response quality and we also introduce a realistic tradeoff between utility and the skill compositions along with further analysis on this more realistic setting.
>
> With weighted intents, the game remains **tractable at the structural level**: the attacker’s best-response still selects skill compositions that minimize the effective defense accuracy $a_{i,s}​$ and the defender still faces a constrained resource allocation problem across intent–skill pairs. While closed-form expressions such as Theorem 3.2 may no longer exist due to the added heterogeneity in  $w(i)$ and  $u(i,y) $, the equilibrium reduces to a **weighted resource allocation problem**.
>
> Incorporating weighted intents does not change the core qualitative conclusions of our framework. The attacker continues to exploit the weakest intent–skill combinations, and the defender must allocate limited capacity under combinatorial scaling of the skill space. The primary effect of weighting is to **shift the defender’s allocation toward high-severity or high-utility intents**, yielding a more risk-sensitive equilibrium. Notably, the attacker’s combinatorial advantage driven by the size of the skill composition space remains unchanged, indicating that our main insights are robust beyond the simplified setting.
>
> We will revise the paper to clarify that Assumption 2.2 serves as a **conservative case**, while the more realistic formulation in Appendix B captures heterogeneous utilities.
>
> We appreciate the reviewer’s insightful questions and suggestions, particularly regarding the interpretation of the minimax objective and the role of intent weighting. We hope our clarifications address these points. Please let us know if there are any further questions or aspects we can clarify or elaborate further.

---

> > ### Author Rebuttal · Reviewer_SWHC · 2026-04-03
> >
> > I thank the writers for their elaboration on the connection between the theoretical framework and the practice of LLM jailbreaking. While I find that some important aspects of practical jailbreaking are lacking in the theory (e.g. intent weight - varying severity of different intents), I am overall supportive of the work, thus I will maintain my score.

---

> > > ### Author Response · Authors · 2026-04-07
> > >
> > > We sincerely thank the Reviewer SWHC for the positive feedback and for highlighting the importance of **modeling heterogeneous intent severity**.
> > >
> > >
> > > The $w(i)$ in our equation 2 can capture the relative importance (severity) of intent $i$. We extend Theorem 3.2 to a setting with weighted intents:
> > >
> > >
> > >
> > > Under the same conditions as Theorem 3.2, but with the payoff given by
> > > $$
> > > V(i,x,y,D)=w(i)(1-D(x,y)),
> > > $$
> > > The equilibrium value is
> > > $$
> > > J_w^\star(I) = \sum_{i\in I} p(i) w(i) - \frac{c}{|S^{(k)}|}\max_{i\in I} p(i)w(i).
> > > $$
> > >
> > >
> > >
> > > **Proof:** The result follows from a direct modification of Theorem 3.2. Under the weighted payoff $V(i,x,y,D)=w(i)(1-D(x,y))$, the best-response attacker remains unchanged: for each intent $i$, it selects a skill composition minimizing $a_{i,s}$. Thus, Proposition 3.1 applies with coefficients $p(i)$ replaced by $p(i)w(i)$.
> > >
> > > Reusing the same argument as in Theorem 3.2 for the defender, the optimal allocation within each intent is uniform across skills, and the total budget is assigned to the intent with the largest coefficient $p(i)w(i)$. Substituting this solution yields
> > > $$
> > > J_w^\star(I) = \sum_i p(i)w(i)-\frac{c}{|S^{(k)}|}\max_i p(i)w(i).
> > > $$
> > >
> > >
> > > This is a simple and direct extension showing that incorporating intent severity does not change the core structure of the game: the attacker still exploits weak intent–skill pairs, and the defender still faces a combinatorial allocation problem. Our original formulation can be viewed as a **normalized special case**, where $w(i)$ is absorbed into $p(i)$.
> > >
> > > The main effect of $w(i)$ is to shift the effective importance of each intent from $p(i)$ to $p(i)w(i)$, giving a more realistic interpretation: **(1)** a rare but very severe intent can dominate defense allocation; **(2)** while a common but low-severity intent may deserve less capacity.
> > >
> > > We will clarify this extension in the revision to better connect the theory with practical jailbreak settings.
> > >
> > > We thank the reviewer again for the constructive feedback, which has strengthened the paper. In addition, we provide extensive new empirical evidence: **(1) comparisons with additional methods on the latest frontier LLMs such as GPT 5.4**, and **(2) experiments on substantially larger and more diverse skill sets** and **(3) an additional formal construction and proof for a non-binary abstraction** in response to concerns raised by other reviewers.  We hope the above clarifications and additional evidence adequately address the remaining concerns. We would be deeply grateful if these improvements will **be reflected in the final assessment in light of the extensive new evidence provided**.

---

### Decision · Program_Chairs · 2026-04-30

**Decision:**

Accept (regular)

**Comment:**

After rebuttals, this paper still sits at the border between accept/reject -- two reviewers gave a weak reject (confidence 4 and 5, resp.) and one reviewer gave a weak accept (confidence 3). The reviewers highlighted the paper's "compelling and original framework" (`SWHC`), the "interesting and. . . potentially useful unifying perspective" (`CREG`), and mathematical rigor (`8Yzv`). Indeed, the framing of adversarial jailbreaking as a two-player game does shed light on the problem, and the overall consensus was that this perspective has the potential to shed more light on this important/evolving area of research.

The reviewers also pointed to a number of weaknesses, including the fact that the experiments are relatively out of date (`8Yzv`, `CREG`). While the theory in this paper does provide a nice unifying perspective, it's certainly the case that the models, datasets, and algorithms are all relatively old. This isn't necessarily a problem from a scientific perspective, although the authors do make a few claims about the evolving harmful capabilities of models, which fall a little bit flat given that the most recent model they evaluated was GPT-4 (Mar. 2023). The authors provided some new experiments, which should resolve some of the original concerns in my view.

The other main point of criticism was reliance on assumptions that _may not_ hold in practice (`SWHC`, `CREG`). The authors [argued](https://openreview.net/forum?id=s6Vno2TwLJ&noteId=KHSsCVUghE) (very thoughtfully, in my opinion) that while the assumptions may not hold in practice, they still retain some relevance. And beyond this, the authors also look into a relaxed set of instructions. Two things occur to me here: (a) from the perspective of doing good science, it's perfectly reasonable to start with idealized assumptions; and (b) the flexibility/nuance with which the authors adapted their work genuinely improved the paper.

All this being said, I think that the main concerns raised were actually addressed by the authors. We should not, in my view, reject papers simply because their assumptions do not line up one-to-one with reality, especially given that the assumptions seem realistic enough to be of some practical use here. Thus, provided that the authors are willing to add their new experiments in more up-to-date settings to the paper, I support accepting this paper.